

# Physics-informed machine learning for understanding rock moisture dynamics in a sandstone cave

Kai-Gao Ouyang[1], Xiao-Wei Jiang[1,2*], Gang Mei[3], Hong-Bin Yan[4],

Ran Niu[1], Li Wan[1], Yijian Zeng[5]

1. MWR Key Laboratory of Groundwater Conservation, China University of Geosciences, Beijing, China

2. MNR Key Laboratory of Shallow Geothermal Energy, China University of Geosciences, Beijing, China

3. School of Engineering and Technology, China University of Geosciences, Beijing, China

4. Yungang Research Institute, Datong, Shanxi, China

5. Department of Water Resources, ITC Faculty of Geo-Information Science and Earth Observation, University of
     Twente, Enschede, the Netherlands

***Correspondence:** Xiao-Wei Jiang (jxw@cugb.edu.cn)

15                                                    **Abstract**

Rock moisture, which is considered as a hidden component of the terrestrial hydrological cycle, has received

little attention. In this study, the frequency-domain reflectometry (FDR) is used to obtain fluctuating rock water

content in a sandstone cave of the Yungang Grottoes, China. We identified two major cycles of rock moisture

addition and depletion, one in the summer and the other in the winter. By using the LSTM (Long Short-Term

Memory) network and the SHAP (SHapley Additive exPlanations) method, relative humidity, air temperature and

wall temperature are found to have contributions to rock moisture in the summer. By using vapor concentration

and the difference between dew point temperature and wall temperature as two input variables of the LSTM

network, the predicted rock water content has a very good agreement with the measured rock water content, with

the Nash–Sutcliffe efficiency coefficient (NSE) being as high as 0.978. Because the two new input variables are

factors directly controlling vapor condensation, they provide informative priors to the deep learning model and

improved prediction performance. After identifying the causal factors of rock water content fluctuations, we also

identified the mechanism controlling the multi diurnal vapor condensation. The increased vapor concentration



accompanying a precipitation event leads to transport of water vapor into rock pores, which is subsequently

adsorbed onto the surface of rock pores and then condensed into liquid water. With the aid of the deep learning

model, this study increases understanding of sources of water in caves, which would contribute to future strategies

of alleviating weathering in caves.

# 1 Introduction

Water movement in the unsaturated zone is a fundamental component of the hydrologic cycle regulating the

atmosphere, the hydrosphere and the lithosphere (Arora et al., 2019; Brubaker and Entekhabi, 1996; Lu and Likos,

2004; Tindall et al., 1998). Although there are abundant studies on water movement in various scales of unsaturated

soils (Larson et al., 2022; Schoups et al., 2005; Vereecken et al., 2014; Vinnikov et al., 1996; Yu et al., 2016), much

less attention has been paid to water in unsaturated rocks. In a recent study, Rempe and Dietrich (2018) defined

water stored in unsaturated rocks as rock moisture and pointed out that rock moisture is a hidden component of the

terrestrial hydrologic cycle critical to ecosystems and weathering processes. Due to the ubiquitous occurrence of

precipitation infiltration through unsaturated rocks, infiltrating precipitation was found to be the main source of

rock moisture (Rempe et al., 2018; Sass, 2005). In fact, as early as in the fourth century BC, Aristotle (384-322 BC)

hypothesized that atmospheric water vapor could penetrate into rocks in caves with low temperature and condense

into liquid water (after Meinzer, 1934). Due to occurrence of hidden water in the form of rock moisture, many stone

heritages inside caves have suffered from weathering (Auler and Smart, 2004; Camuffo, 1998; de Freitas et al.,

2006; Guerriera et al., 2019; Linan et al., 2021). However, up to now, there is no observations of rock moisture in

caves, which hampers a comprehensive understanding of the source and control factors of rock moisture.

By using such techniques as downhole neutron probe (Rempe and Dietrich, 2018), time domain reflectometry

(TDR) (Salve et al., 2012) and nuclear magnetic resonance (NMR) (Schmidt and Rempe, 2020), the responses of

rock moisture to precipitation were identified in some small catchments. However, these devices, which are usually

long in length (the lengths of TDR and neutron probe are larger than 30 mm) or large in diameter (the diameter of

NMR is around 70 mm), are not suitable to be used in stone heritages. The frequency domain reflectometry (FDR),

which has the advantage of small in volume (the length is less than 60 mm), has been widely used to characterize

the temporal variability of soil moisture (Irmak and Irmak, 2005; Li et al., 2020; Xie et al., 2021; Zhang et al.,

2019). Note that both TDR and FDR measure the dielectric constant of porous media, and FDR is less influenced

by temperature (Zhu et al., 2019). In this study, we attempt to use the FDR for monitoring rock moisture in the field



for the first time and identify the atmospheric conditions controlling rock moisture in caves not exposed to rain.

Establishing the cause-and-effect relationship between rock moisture and various atmospheric conditions is a feasible approach to identify the source of rock moisture in caves and reveal mechanisms controlling rock

moisture fluctuations. Machine learning has the ability to acquire knowledge and establish the complicated nonlinear relationship between variables in a vast domain (Chen et al., 2019a; Jumin et al., 2020). Although machine learning models have the ability of high accuracy prediction, they are notorious for being a black-box model. Lundberg and Lee (2017) proposed the SHAP (SHapley Additive exPlanations) values as a unified measure of feature importance, which led to a combination of accuracy and interpretability of predictions by machine

learning models. In almost all applications of machine learning in the field of hydrology, the directly measured meteorological factors like precipitation, temperature, radiation, humidity and wind speed are used as input variables (e.g., Barzegar et al., 2017; Fang et al., 2017; Gao et al., 2020; Lees et al., 2021; Liu et al., 2022; Xiang et al., 2019; Zhao et al., 2022). In fact, using prior knowledge stemming from physical or mathematical understanding as model inputs could improve the performance of a machine learning algorithm, which is called

physics-informed learning (Karniadakis et al., 2021).

In this study, the classic Long Short-Term Memory (LSTM) network, which is deep learning model, is combined with the SHAP values to predict rock moisture and evaluate the relative importance of four directly measured variables (precipitation, relative humidity, air temperature and wall temperature). After excluding the possible control by precipitation, based on the physics controlling vapor condensation, two new variables derived

from relative humidity, air temperature and wall temperature are used as inputs of the LSTM network, which not only leads to improved prediction performance, but also improve understanding of source of water in caves.

## 2 Study site and field monitoring

### 2.1. Study site

The Yungang Grottoes (40°07' N, 113°08' E), which are located in Datong, Shanxi Province, China, were

declared World Heritage Site by the UNESCO in 2001 (http://whc.unesco.org/en/list/1039). According to meteorological data in recent 20 years in the Datong city (data from The China Meteorological Data Service Center, http: //data.cma.cn/en), the site has a semi-arid climate, with an annual average precipitation of 393 mm and an annual average pan evaporation of 1243 mm. The precipitation in the rainy season from June to September accounts for 73% of the annual precipitation. The annual average temperature is 8 ℃, the average temperature in summer is

20.3 ℃, and the average temperature in winter is -8.2 ℃.

Most statues in the Yungang Grottoes were carved in sandstone caves (Fig. 1a). In the summer, water droplets
with planar distribution can be occasionally observed on the walls of some caves (Fig. 1b). Although no water
droplets occur in other sandstone walls, by absorbing water, the high rock moisture leads to slight changes in the
color of some walls. These different forms of water are responsible for weathering of the statues, however, the
sources of these forms of water remains controversial. Previous studies suggested that the possible sources of water
in caves include infiltrating precipitation through the overlying thick unsaturated zone (Wang et al., 2012) and
condensation of water vapor onto walls (Cao et al., 2005; Huang, 2008; Huang, 2010).

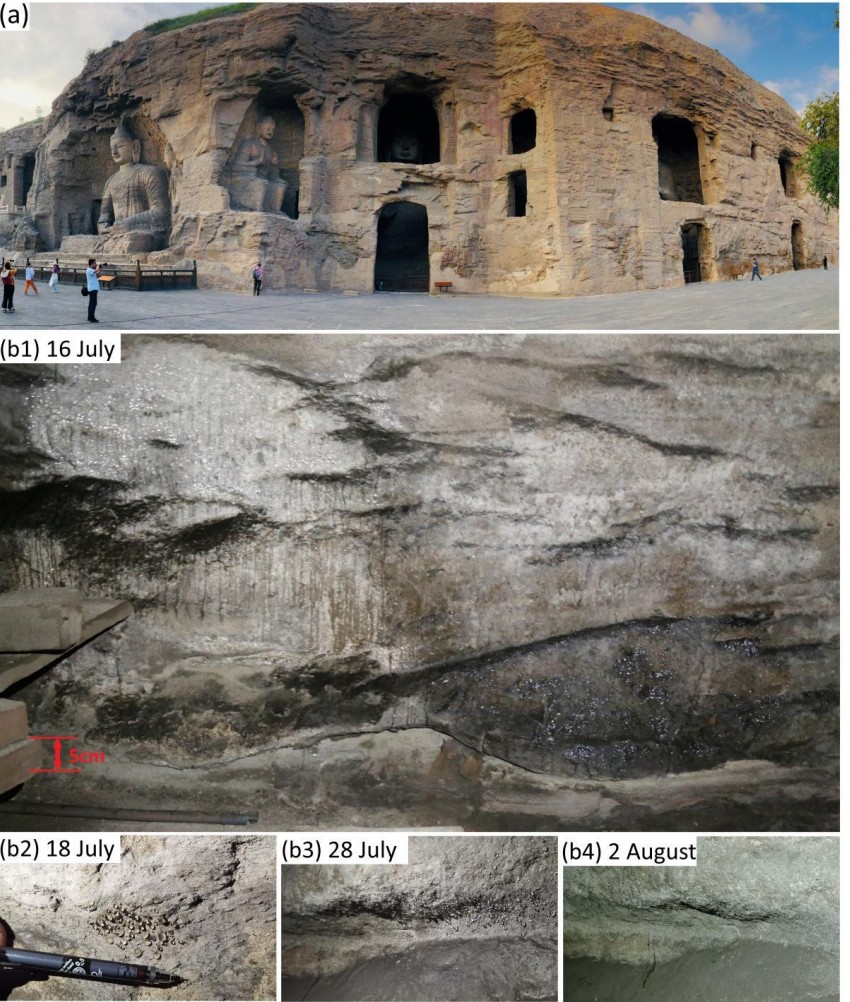

Figure 1. (a) Some caves and statues in the Yungang Grottoes; (b) The occurrence (b1-b3) and disappearance (b4)
of water droplets in Cave #5 of the Yungang Grottoes in the summer of 2021.



### 2.2. Monitoring of rock moisture and atmospheric conditions

To monitor variations of rock moisture in the shallow part of a cave wall, a FDR-based ECH2O EC-5 sensor (produced by DECAGON, USA) was installed at 3-8 cm inside the north wall of Cave #9 (Fig. 2). Due to the difficulty of calibrating the actual water content in the field, the apparent rock water content measured by the sensor is directly used in the current study. As reported in previous experimental studies (Mollo and Greco, 2011; Sakaki and Rajaram, 2005), there is a good linear relationship between actual rock water content and rock moisture transformed from dielectric constant. Because the purpose of the current study to identify the source of rock moisture addition by establishing the relationship between rock moisture and atmospheric conditions, the relative changes of rock water content is more important than the actual value of rock water content. Therefore, direct use of apparent rock water content does not influence the results of our study, and the apparent rock water content is directly called rock water content.

Air temperature ($T_a$) and air relative humidity ($RH$) are simultaneously monitored near the monitoring site of rock moisture (Fig. 2). The wall temperature is also monitored to analyze whether the wall meets the condition for condensation. Moreover, hourly precipitation is available from a meteorological station outside the cave.

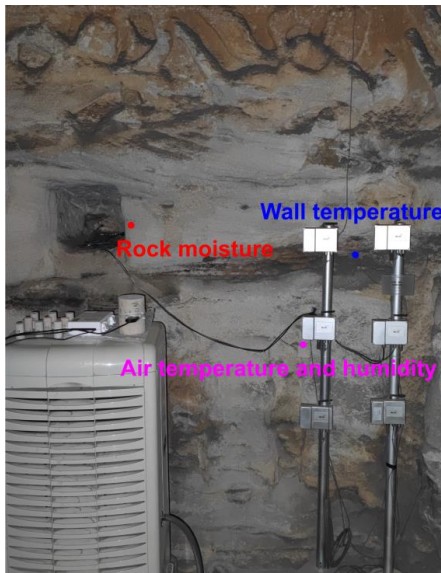

Figure 2. A photo showing the arrangements of sensors for rock moisture, wall temperature, air temperature and relative humidity in the north wall of Cave #9



## 3 Methods

### 3.1 Long short-term memory (LSTM) network

The LSTM network is an improved variant of the conventional Recurrent Neural Network (RNN) and has

the same fundamental framework as the conventional RNN. Therefore, we first give a brief introduction to the

conventional RNN. As shown in Fig. 3, in the RNN, there is an input layer, a hidden layer and an output layer. $x_t$ is

the input vector at time step $t$, $h_t$ is the hidden state at time step $t$ determined by both the input vector $x_t$ at time step

$t$ and the hidden state ($h_{t-1}$) at time step $t$-1 (Zhao et al., 2017), and $op_t$ is the output of the RNN at time step $t$.

Mathematically, the relationship between the three layers can be written as:

$$h_t = tanh \, (U x_t + W h_{t-1}) \tag{1}$$

$$op_t = V h_t \tag{2}$$

where *tanh* is the activation function which means the hyperbolic tangent performs nonlinear transformations of

the inputs, $U$, $W$ and $V$ are the network weight matrices for input-to-hidden, hidden-to-hidden and hidden-to-output

connections, respectively, $b_o$ and $b_h$ are bias vectors. In this study, the open-source framework TensorFlow (version

1.14.0) written in Python 3.7.6 is used to build and train the LSTM model.

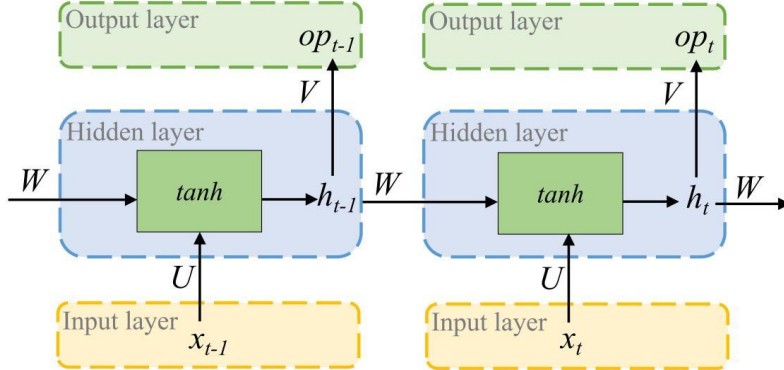

Figure 3. The structure of the Recurrent Neural Network (RNN) (Modified from Hopfield, 1982).

When the time sequence is too long, the RNN is not sufficiently efficient to learn because of vanishing

gradient and exploding gradient problems (Bengio, 1994; Hochreiter and Schmidhuber, 1997). To solve such

problems, Hochreiter and Schmidhuber (1997) proposed an improved variant of RNN whose hidden layer can

capture the correlation within time series in both short and long term. This improved variant was named the LSTM



network, and its hidden layer was called the LSTM cell. As shown in Fig. 4, a LSTM cell has three gates, including

a forget gate, an input gate and an output gate, to maintain and adjust its cell state $c_t$ (also called memory) and

hidden state $h_t$. The forget gate determines what information should be discarded from the cell state, the input gate

specifies what new information need be stored in the cell state, and the output gate determines what information

from the cell state $c_t$ should be passed to the next step (Gao et al., 2020; Fischer and Krauss, 2018; Lipton et al.,

2015). Note that in the input gate, the cell state is determined simultaneously by the the sigmoid activation function

which determines the weights of the information and the *tanh* activation function which eliminates the bias of the

network, the latter of which is termed abstract cell state.

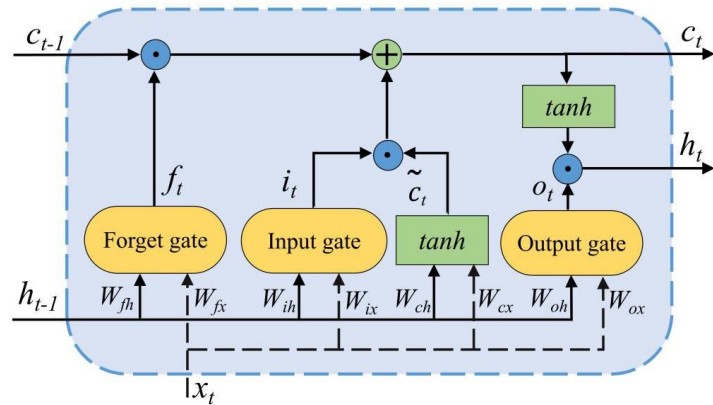

Figure 4. The structure of the LSTM cell (Modified from Hochreiter and Schmidhuber, 1997).

The formulas of the three gates, abstract cell state, cell state and hidden state in the LSTM cell are shown as

follows:

(forget gate)             $$f_t = \sigma\left(W_{fx}x_t + W_{fh}h_{t-1} + b_f\right) \tag{3}$$

(input gate)             $$i_t = \sigma\left(W_{ix}x_t + W_{ih1}h_t\right) \tag{4}$$

(output gate)            $$o_t = \sigma\left(W_{ox}x_t + W_{oh}h_{t-1} + b_o\right) \tag{5}$$

(abstract cell state)     $$\tilde{c}_t = tanh\left(W_{cx}x_t + W_{ch}h_{t-1} + b_c\right) \tag{6}$$

(cell state)             $$c_t = f_t \odot c_{t1} + i_t \odot \tilde{c}_t \tag{7}$$

(hidden state)           $$h_t = o_t \ tanh \tag{8}$$

where $i_t$, $f_t$ and $o_t$ are the vectors of input, forget and output gates at time step $t$, respectively, all of which have the





same sizes as $c_t$ and $h_t$, σ is the logistic sigmoidal activation function, $\tilde{c}_t$ is the vector of abstract cell state at time

step $t$, ⊙ is element wise multiplication of two vectors. Similar to RNN, $W_{ix}$, $W_{ih}$, $W_{ox}$, $W_{oh}$, $W_{cx}$ and $W_{ch}$ are the

matrices for different connections in the network, $b_i$, $b_o$ and $b_c$ are bias vectors. In this study, the open-source

framework TensorFlow (version 1.14.0) written in Python 3.7.6 is used to build and train the LSTM model.

Although purely data-driven models may have a high accuracy of prediction, most machine learning models

cannot extract interpretable information and knowledge from this data deluge (Karniadakis et al., 2021). The

performance of machine learning models could be improved through providing physical prior knowledge. In this

study, we simply integrate physics controlling vapor condensation into the input variables to improve the

performance of the LSTM model.

### 3.2 Model interpretation and evaluation

To interpret the performance of a machine model, Lundberg and Lee (2017) proposed the SHAP (SHapley

Additive exPlanations) explanation method, which is based on game theory (Štrumbelj and Kononenko, 2014).

The Shapely value of every input variable represents the contribution of it on the prediction, and the importance of

each input variable is clarified by comparing model performances with and without it. The formula for calculation

of the Shapely value is

$$\varnothing_i = \sum_{S \subseteq F \setminus i} \frac{|S|! \, |F| - |S| - 1\,!}{n!} \left[ v\,(S \cup i) - v\,(S) \right]$$

(9)

where $\phi_i$ is the contribution of variable $i$; $F$ is the set of all input variables; $v(S \cup \{i\})$ is the result of a model trained

with the variable $i$, and $v(S)$ is the result without the variable $i$, so the difference between them represents the effect

of feature $i$ on the model prediction. This method requires retraining the model on all feature subsets $S \subseteq F$ (Shapley,

1953).

To assess the accuracy of prediction by the LSTM network, we use the statistical metrics of Nash–Sutcliffe

efficiency coefficient (NSE), mean absolute error (MAE) and root mean squared error (RMSE), all of which are

widely used in the literature. NSE is the ratio of the sum of the squares of the regression to the total sum of the

squares, which reflects the linear fit between the predictions and observations. The closer the value is to 1, the

better the linear fit. The expression of NSE is

$$NSE = 1 - \frac{\sum_{i=1}^{N} \left( y_{Pred} - y \right)^2}{\sum_{i=1}^{N} \left( \bar{y} - y \right)^2}$$

(10)

where N is the number of data, and $y_{Pred}$, $y$, and $\bar{y}$ are the predicted, observed, and mean observed value, respectively.





MAE is the mean of the distance between the predicted and the observations, whereas RMSE is the square root of the mean of the square of the deviation between the predicted and the observations. The expressions of MAE and RMSE are

$$MAE = \frac{1}{N} \sum_{i=1}^{N} \left| y_{Pred} \right| - \left| y \right| \tag{11}$$

$$RMSE = \sqrt{\frac{1}{N} \sum_{i=1}^{N} \left( \left| y_{Pred} \right| - \left| y \right| \right)^2} \tag{12}$$


## 4 Results and discussion

### 4.1 The seasonal variation of rock moisture and atmospheric conditions

In the north wall of Cave #9, although there is no obvious occurrence of liquid water throughout the year, there is a clear trend of seasonal variation in rock water content (Fig. 5). From April to May, the rock water content

is relatively stable and maintains at around 0.013 cm$^3$/ cm$^3$. From June to September, which correspond to the rainy season with high relative humidity and high air temperature, there is a cycle of significant addition and depletion of rock moisture. From October to December, there is a trend of gradual decrease in rock water content. The cycles of precipitation, relative humidity, air temperature and wall temperature from spring to early winter have quite similar trends as the cycle of rock water content, indicating that they are possible environmental conditions leading

to the fluctuating rock water content.

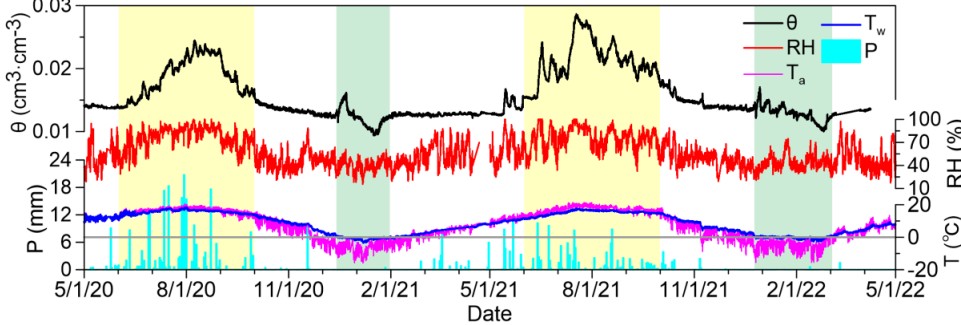

Figure 5. The temporal variations of rock water content ($\theta$), air relative humidity ($RH$), air temperature ($T_a$), precipitation ($P$) and wall temperature ($T_w$). The two periods in yellow correspond to the summer period with high temperature and high humidity, whereas the two periods in green correspond to the fluctuation of rock moisture

caused by freezing-thawing.





In the summer of 2021, the rock water content has a sharp increase since 9 July, reaches a maximum value equaling 0.029 $cm^3/cm^3$ on 17 July, and is maintained at relatively high values until 28 July. The high rock water content indicates that there are atmospheric conditions responsible for water infiltration or condensation of water vapor. Note that this period corresponds to the period with occurrence of water droplets in Cave #5 as shown in Fig. 1b1-b3. Although there is no water droplet in Cave #9, the color of the sandstone changes slightly, indicating that a slight change in the color of sandstone is a result of the fluctuating rock water content. Therefore, we believe that the rock water content measured by FDR reflects the actual change of water content in the rock.

From mid December in 2020 to the end of January in 2021, and from late December 2021 to the end of February in 2022, there are also significant fluctuations of rock water content (Fig. 5). This pattern of fluctuating rock water content is a direct of freezing-thawing, which can be confirmed by the negative wall temperature. At the beginning of the freezing-thawing cycle, there is a trend of increasing rock water content around the sensor due to freezing-induced liquid water migration towards the wall surface with the lowest temperature. By the end of the freezing period, the rock water content reaches a minimum value of the year because most liquid water has been transformed into ice. In the two years, the minimum liquid water content is 0.009 $cm^3/cm^3$ (on 16 January 2021) and 0.010 $cm^3/cm^3$ (on 25 February 2022), respectively. In the thawing stage, there is a trend of increasing liquid water content.

The pattern of freezing-thawing-induced rock water content fluctuations is similar to that of freezing-thawing-induced soil water content fluctuations (Deprez et al., 2020; Matsuoka and Murton, 2008; Sun and Scherer, 2010; Xie et al., 2021; Yu et al., 2018), demonstrating that the FDR technique is very sensitive to liquid water content in sandstone and is suitable to measure rock moisture. The fluctuating rock water content during the freezing-thawing cycle also has implications for understanding weathering processes. The decreasing low liquid water content induced by freezing indicates that rock moisture in spring and autumn belongs to movable bound water that could be responsible for chemical weathering. Moreover, the freezing of accumulated liquid water near the wall surface might cause physical weathering.

## 4.2 The performance of LSTM model with two different schemes

In the rainy season, as we pointed out in 4.1, precipitation, relative humidity, air temperature and wall temperature have quite similar trends of seasonal variation as apparent rock moisture. Apparently, they are all possible factors determining the fluctuating apparent rock moisture. Therefore, in 4.2.1, we first use all of them as input variables (scheme #1) of the LSTM model to predict rock water content, and use the SHAP values to evaluate



the contribution of each input variable. After excluding precipitation whose mean |SHAP value| equals 0, in 4.2.2, we use two new parameters (vapor concentration, dew point temperature minus wall temperature) calculated from relative humidity, air temperature and wall temperature as input variables (scheme #2) of the LSTM model to predict apparent rock moisture.

Deep learning models require a large amount of data for training, as well as data sets with a longtime span to ensure the mastery of complete data features. Because the period from 1 June to 1 October has the most significant trends of rock moisture addition and depletion, the hourly data during this period in the year 2020 are used to construct the training set, whereas the hourly data in the year 2021 are used to construct the test set.

### 4.2.1 The predicted results based on directly monitored variables

By using relative humidity, air temperature, precipitation and wall temperature as model input variables, there is a fairly good match between the predicted and measured rock water content, with similar patterns of fluctuations (Fig. 6a). Although there is obvious underestimation of rock water content in mid and late July, and slight underestimation or overestimation in other months, the NSE is as high as 0.958, indicating that the fluctuating relative humidity, air temperature, precipitation and wall temperature can capture the major patterns of fluctuating

rock water content.

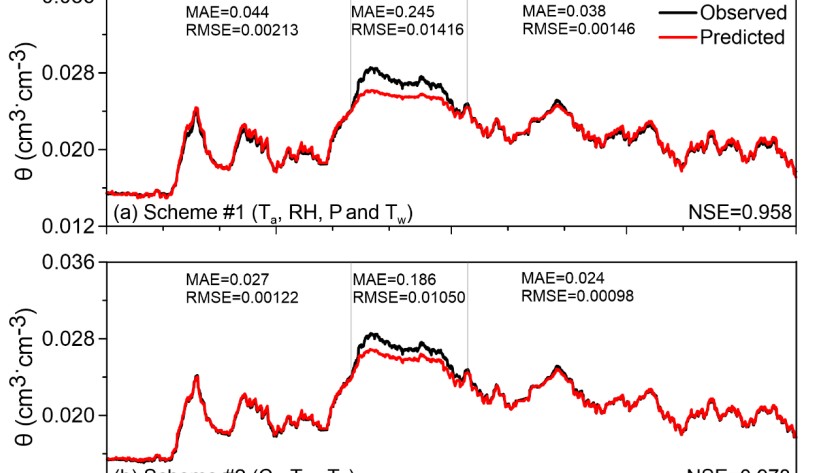

Figure 6. The predicted and measured rock water content obtained by two schemes. (a) Scheme #1 using four directly measured variables; (b) Scheme #2 using two calculated variables controlling vapor condensation. Also shown are NSE of the whole time series, MAE and RMSE of three different stages.



Fig. 7a shows the mean absolute SHAP value of each input variable, which represents the relative importance of each variable for the prediction. The mean absolute SHAP values of air relative humidity, air temperature, wall temperature, rock water content at previous time step and precipitation are 0.0087, 0.0032, 0.0018, 0.0004 and 0, respectively. Therefore, precipitation has no direct contribution to rock moisture in caves, and we infer that vapor condensation should be the source of rock moisture in caves.

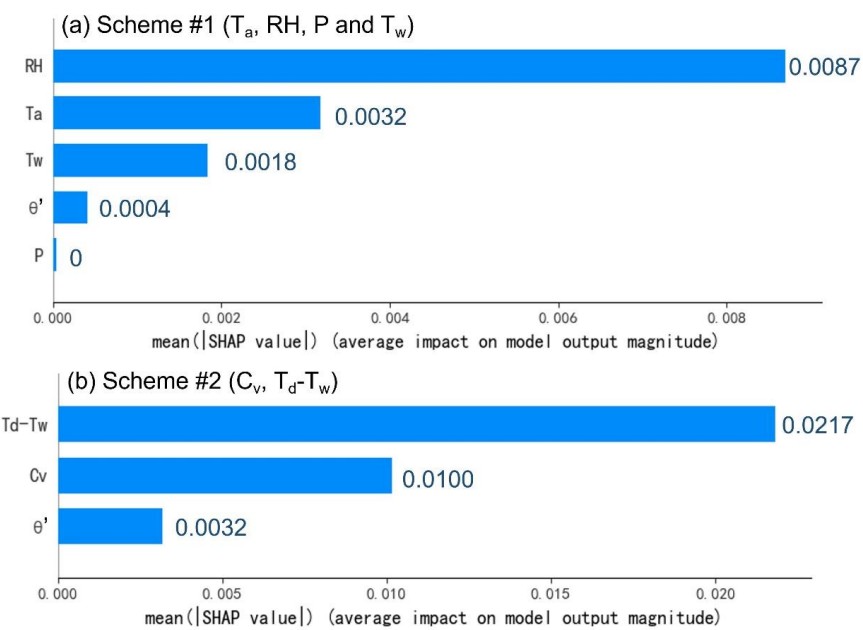

Figure 7. The relative importance of each input variable in the two schemes with different input variables. (a) Scheme #1 using four directly measured variables; (b) Scheme #2 using two calculated variables controlling vapor condensation. Note that rock water content at previous step ($\theta'$) also has contribution to prediction.


**4.2.2 The prediction results based on variables controlling vapor condensation**

Based on the SHAP values of scheme #1, precipitation can be excluded as an input variable for the LSTM network. Among the three directly monitored variables that have contributions to rock water content, air relative humidity and air temperature determines the vapor concentration and the dew point temperature (Nguyen et al., 2014), and whether the wall temperature is below the dew point temperature determines whether vapor

condensation could occur (Fernández-Cortés et al., 2006; Gabrovšek et al., 2010; Li et al., 2021). Because water vapor is the direct source of condensation water and whether the wall temperature is below the dew point





temperature is the precondition of condensation, we use vapor concentration and dew point temperature minus wall temperature as two input variables, both of which are direct variables controlling vapor condensation.

Vapor concentration and dew point temperature are both functions of actual vapor pressure, which is

determined by air temperature, saturated vapor pressure and relative humidity. For air with a temperature of $T$ (K), the formula for calculating saturated vapor pressure and actual vapor pressure are (Lawrence, 2002; Lu, 2004):

$$u_{v,sat} = 0.611 \exp\left(17.27 \frac{T-273.2}{T-36}\right) \tag{13}$$

$$u_v = u_{v,sat} \cdot RH \tag{14}$$

where $u_{v,sat}$ is the saturation vapor pressure (kPa), $RH$ is the relative humidity of air (%), and $u_v$ is the actual

vapor pressure (kPa). After obtaining $u_v$, the vapor concentration, $C_v$ (g/m$^3$), and the dew point temperature, $T_d$ (K), can be calculated as (Lu, 2004)

$$C_v = 217 \cdot \frac{u_v}{T-273.15} \tag{15}$$

$$T_d = \frac{36\ln(u_v) - 4700}{\ln(u_v) - 16.78} \tag{16}$$

As indicated in Equation 16, a higher water vapor content in the air, $u_v$, corresponds to a higher dew point

temperature, thus a higher possibility of condensation at the wall. Fig. 8 shows that the patterns of fluctuating rock water content, vapor concentration and difference between dew point temperature and wall temperature (denoted as $T_d$-$T_w$ hereafter) in the whole non-freezing period are quite similar. Moreover, we find the period with a positive $T_d$-$T_w$ has a good correspondence with the period with a high level of rock moisture.

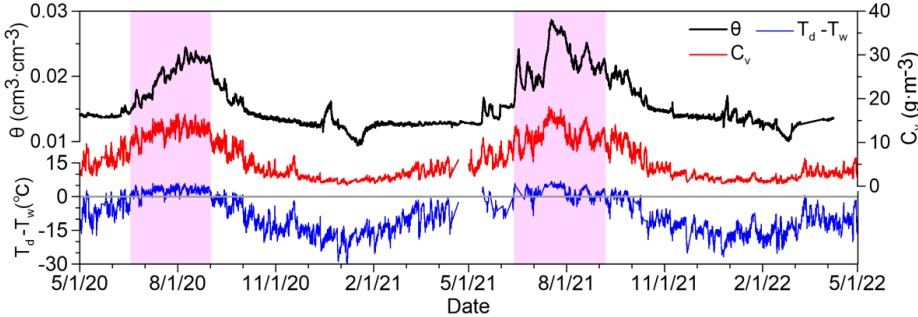

Figure 8. The fluctuating apparent rock moisture ($\theta$), vapor concentration ($C_v$), and the difference between dew point temperature and wall temperature ($T_d$-$T_w$). The zones in purple represents the periods with positive $T_d$-$T_w$ and high rock moisture.



In scheme #2, by using the two new variables as input of the LSTM model, the mean absolute SHAP values of $T_d$-$T_w$ and vapor concentration are 0.0217 and 0.0100, respectively (Fig. 7b), indicating that both variables have

significant contributions to rock moisture. Moreover, the NSE of predicted rock moisture is further increased to

0.978 (Fig. 6b). Although the prediction still underestimates rock water content from mid July to the end of July,

the MAE reduced from 0.245 in scheme #1 to 0.186 in scheme #2, and RMSE reduced from 0.01416 in scheme #1

to 0.01050 in scheme #2. In the other two time durations, the MAE and RMSE of scheme #2 also decrease obviously.

Therefore, scheme #2 has much better performance of prediction, showing that using physics-informed variables

would improve accuracy of prediction.

### 4.3 The mechanism of water vapor condensation

As we illustrated in 4.2, precipitation is not directly responsible for rock moisture fluctuations, but other

atmospheric conditions controlling vapor concentration and the condition of vapor condensation are directly

responsible for rock moisture fluctuations in the cave. In fact, vapor concentration fluctuations are more or less

related to precipitation events. As shown in Fig. 9, the vapor concentration usually begins to rise prior to the

occurrence of a precipitation event, and declines under the control of solar radiation after a precipitation event.

Under the control of convection and diffusion, the increased water vapor in the air could be transported into

porous media. When the sandstone is dry, water vapor can be adsorbed onto the surface of the rock pores, forming

an adsorbed layer as thin water film; as curved menisci begin to form under increasing relative humidity, capillary

condensation occurs in the rock pores (Broekhoff, 1969; Lu and Likos, 2004; Xu et al., 1998). Both adsorption and

capillary condensation would lead to rock moisture addition. As shown in Fig. 9, in the summer of 2021, there are

10 stages with obvious rock moisture additions. In the majority of the 10 stages, there are lagged responses of rock

moisture additions to rising vapor concentration in the air, probably due to time required for vapor convection and

diffusion.

Among the 10 stages, the magnitude of rock moisture addition is controlled by $T_d$-$T_w$. In stages IV, V, VII,

VIII, IX and X, because the dew point temperature seldom exceeds the wall temperature, the magnitudes of rock

moisture additions are relatively small. In stages I, II, III and VI, there are long durations with dew point

temperature being higher than the wall temperature, causing large magnitudes of rock moisture addition. However,

at the beginning of these four stages, even if dew point temperature is still lower than the wall temperature,

increasing vapor concentration has resulted in rock moisture addition. Therefore, although a negative $T_d$-$T_w$ does

not exclude the possible occurrence of capillary condensation, a positive $T_d$-$T_w$ does promote capillary





condensation.

Following the 10 stages of rock moisture additions, we find rock moisture depletion is very sensitive to decreasing vapor concentration. Moreover, in stage III with very high rock water content, a slight decrease in vapor

concentration results in a slight decrease in rock water content. Therefore, we believe that rock water content measured by the FDR technique is sensitive enough to fluctuating vapor concentrations and can be applied in future rock moisture monitoring in other settings.

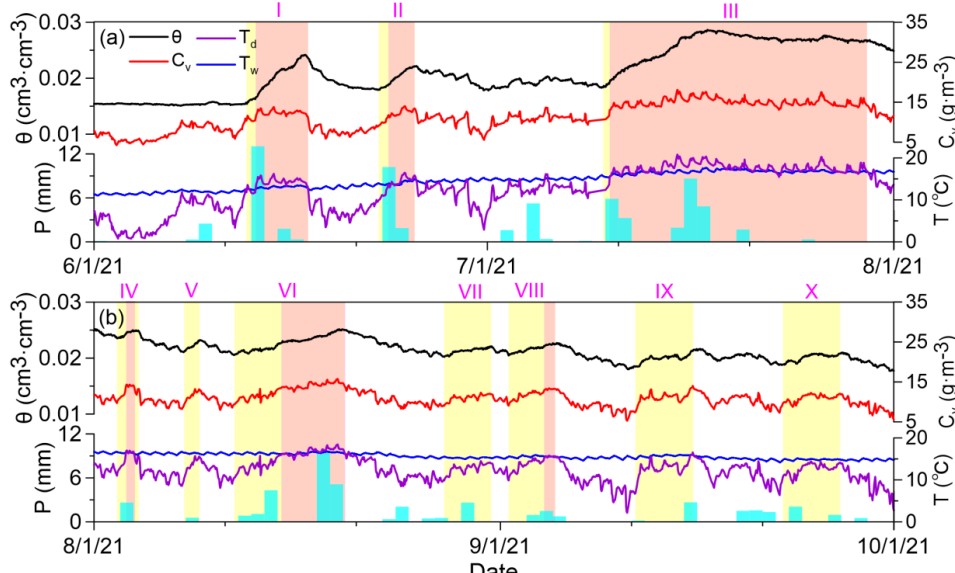

Figure. 9 Plots showing the responses of rock water content ($\theta$) to precipitation ($P$), vapor concentration ($C_v$), dew

point temperature ($T_d$) and wall temperature ($T_w$) in the summer of 2021. The zones in yellow have increasing rock water content, and zones in red have increasing rock water content as well as positive Td-Tw.

## 5 Conclusion

The source of water in the sandstones caves in the Yungang Grottoes responsible for weathering was a long-

standing unresolved scientific question. In this study, we use the FDR to monitor the rock moisture in a cave, which shows clear rock moisture addition-depletions cycles due to various controlling mechanisms. By using relative humidity, air temperature, precipitation and wall temperature as the input variables of the LSTM network, the predicted rock moisture well reproduced the pattern of monitored rock moisture fluctuations. Moreover, we find that precipitation has no contribution, but all other three variables have contribution to the fluctuating rock moisture.



Because relative humidity, air temperature and wall temperature belong to factors controlling vapor condensation, this scheme of deep learning reveals that vapor condensation instead of precipitation is the source of rock moisture in the cave.

By calculating vapor concentration and dew point temperature from air temperature and relative humidity, we proposed two new variables, vapor concentration and the difference between dew point temperature and wall
temperature as input variables of the LSTM network. Because the two variables are direct control factors of vapor condensation, this scheme leads to a much better accuracy of prediction, confirming that rock moisture in the cave is derived from vapor condensation. We also analyzed how precipitation events control vapor concentration, thus indirectly control vapor condensation inside the rock.

By examining the sensitiveness of rock moisture to vapor concentration and the difference between dew
point temperature and wall temperature, this study demonstrates for the first time that the FDR technique is an effective means for monitoring rock moisture. By using variables directly controlling vapor condensation as the input variables of the LSTM model, this study shows that "physics-informed" deep learning can improve prediction performance. Moreover, by identifying how vapor condensation controls rock moisture and occasional occurrence of water droplets in the study area, this study contributes to the understanding of sources of water in caves, which
is important in providing scientific-based proofs to propose future strategies for alleviating weathering of stone heritages.

**Code availability.**

All code used to generate, train, and test the models are available in a dedicated Zenodo repository:
https://doi.org/10.5281/zenodo.7382827.

**Data availability.**

The data of rock water content and atmospheric conditions at the field site (from 1 May 2020 to 1 May 2022) are available in a dedicated Zenodo repository: https://doi.org/10.5281/zenodo.7382895.

**Author contributions.**

All authors were involved in interpretating data provided by HY. XJ developed the initial idea of the current study, and KO run the models with the help of GM. KO and XJ wrote the manuscript with contributions from GM and YZ.



**Competing interests.**

The authors declare that they have no conflict of interest.

**Financial support.**

This study was funded by the National Key R&D Program of China through grant number 2019YFC1520500 and Shanxi Cultural Relics Bureau through contract number 208141400237.

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
