# Peer review of "Physics-informed machine learning for understanding rock moisture dynamics in a sandstone cave"

_Hydrology and Earth System Sciences, 2022_

## Author Comment (AC1)

**Response to Reviewer #1**

This paper investigates the dynamics of rock moisture in a cultural heritage site in China. Based on an analysis of soil moisture sensor measurements, the main outcome of the study is that wetting of the rocks during the summer period is mainly caused by water absorption from the water vapor in the atmosphere. Also a wetting and drying cycle is observed during winter due to freezing and thawing. The interpretation of the results is based on sensitivity analyses of a trained LSTM and an LSTM trained on a set of available direct measurements is compared with an LSTM that is trained on variables that are derived from measurements and that are more directly related to condensation processes such as the wall temperature and dew point temperature. The latter LSTM is called a physics-informed LSTM. The LSTMs are trained on a dataset of one year and are then used to predict the other year.

The obtained insights are interesting and could also be of relevance for the management, protection of the heritage site. I would propose that the authors also give some ideas on how these insights could be used for these purposes.

Response: The water source and formation mechanism in rainy season are concluded in this paper, the vapor concentration is identified as the major control factor of rock moisture in caves, rock heritage protection could be alleviated from the steps of reducing water vapor condensation, rather than rainfall infiltration in this way.

There are a few general questions that need to be addressed.

1) The paper is based on a 2-years time series of rock moisture measurements by only one single sensor. What is the value of such a single sensor time series? Can the results be transferred to other locations? I can imagine that one does not want to disturb such heritage sites with a lot of sensors but wouldn't it be possible to find a few locations where sensor measurements would not cause a disturbance? Or would you expect a big difference when you place the sensors in a sand layer that you put in thermal contact

with the wall so that the wall temperature of the sand is similar to that of the rock? Is a 2-years period sufficiently long? I think these are questions that need to be addressed and discussed in the paper.

Response: (1) Thanks for your understanding. We were not permitted to install a lot of sensors to disturb the heritage sites in the Yungang Grottes. By installing one FDR sensor in Cave #9, which was the first trial of using the FDR technique to monitor rock moisture, we identified the source of rock moisture in the cave from air vapor for the first time. The aim of the current paper is to report our monitoring scheme and reveal sources of rock moisture. Although we only have one sensor, we believe that this does not undermine the effectiveness of the monitoring scheme and the understanding of mechanisms controlling rock moisture fluctuation in caves.

In fact, to test the universality of this recognition, in July 2022, we implemented rock moisture monitoring in Cave #4. The rock moisture measured in Cave #4 and Cave #9 during 21 July and 21 September 2022 are shown in Fig. S1. The similar patterns of rock moisture addition and depletion in response to vapor concentration in the two caves confirm our recognition on mechanisms controlling rock moisture fluctuation in caves. Unfortunately, the data in Cave #4 is limited, which is not suitable to be included into the current manuscript.

[Figure]

Figure S1. The comparison of rock moisture and atmospheric conditions between Cave #4 and Cave #9 in the summer period of 2022.

(2) As we shown in Figure S1, rock moisture in the summer of 2022 is also controlled by vapor concentration. However, rock moisture addition and depletion in the summer

of 2022 are not as significant as those in the year 2021. Moreover, there was no obvious occurrence of water droplets in the caves in the summer of 2022. Because we had good photos showing water droplets in the caves in the summer of 2021, we believe that the year of 2021 is suitable for analysis and prediction.

In fact, we have monitoring data since 2019. After comparing prediction by using 2020 for training and by using 2019 and 2020 for training, we find the predicted results differ little (Figure S2). Because the aim of the current paper is to report our monitoring scheme and reveal sources of rock moisture, we prefer to use data of 2020 and 2021.

[Figure]

Figure S2. The comparison of prediction results.

2) The description of the LSTMs is not clear and the reason for also presenting the RNN is not clear since it is not used in the paper.

Response: Thanks for your comments. We will add more detailed description of LSTM in the revision.

The LSTM network was developed based on the RNN network, for clarity of describing the structure of the LSTM network, many previous papers on the LSTM network (Gao et al., 2020; Kratzert et al., 2018; Fang et al., 2021) described the structure of RNN before introducing LSTM, even if the RNN network was not used for prediction in their studies. Therefore, we follow these studies and present the RNN in our study. We will state the reason clearly in the revision.

3) The physics-informed LSTM performed a bit better than the other LSTM but the

improvements are not very impressive. Could you find examples where you would expect a much larger improvement? Even though the match between the measurement and LSTM is impressive, they remain a black box and it is not so clear to me why the LSTMs are needed in this paper. What is their role? Doing a sensitivity analyses, you found that precipitation did not explain the rock moisture dynamics and comparing model fits of physics-informed LSTMs with non-informed LSTMs you found that absorption of vapor is the main process. But, can't this be inferred as well from time series analyses, e.g. covariance, wavelet, analyses? If the purpose is to obtain an understanding of the processes, what are the advantages of using LSTMs in comparison with other more classical methods? Could the parameters or weights that are obtained by training the LSTM be interpreted and used to explain the behavior of the system?

Response: (1) Thanks for acknowledging that our physics-informed LSTM performed better than the non-physics-informed LSTM. Unfortunately, the prediction still underestimates the measurements in July with the highest rock moisture and we cannot figure out a much larger improvement at this stage. We will try to further improve the accuracy of prediction in the future.

(2) We totally agree that a LSTM model capable of high precision prediction is a black box. The combination of the LSTM model and the SHAP (SHapley Additive exPlanations) method leads to not only high accuracy of prediction, but also interpretability of predictions. Therefore, by combining with the SHAP method, the LSTM model is no longer a black box.

We agree that the covariance of two time series can be used to reveal the correlation between the two time series. Table 1 shows the covariance between rock moisture and normalized atmospheric conditions in 2020 and 2021. The results show that each variable has a positive correlation with rock moisture. Vapor concentration ($C_v$) has the largest covariance while rainfall (P) has the lowest covariance. This is consistent with the feature importance obtained by the SHAP method. Although we did not predict rock moisture by using other classical statistical methods, we believe that the LSTM network, which is an emerging deep learning approach, has its unique advantages over other

classical methods when it is combined with the SHAP method.

Table S1. The covariance between rock moisture and atmospheric conditions

|  | $T_a$ | RH | P | $T_w$ | $C_v$ | $T_d$-$T_w$ |
|---|---|---|---|---|---|---|
| 2020 | 0.030 | 0.039 | 0.001 | 0.034 | 0.047 | 0.030 |
| 2021 | 0.043 | 0.034 | 0.001 | 0.036 | 0.044 | 0.022 |

4) I was confused about the winter period. Was it also used to train the LSTM? There is a discussion on the sensor measurements during the winter period but I could not find a discussion on the performance of the LSTMs for this period.

Response: Sorry for our unclear expressions.

We want to clarify that the winter period was not used to train the LSTM and was not predicted by the LSTM model. In line 235, we pointed out that "Because the period from 1 June to 1 October has the most significant trends of rock moisture addition and depletion, the hourly data during this period in the year 2020 are used to construct the training set, whereas the hourly data in the year 2021 are used to construct the test set".

Abstract:

There were a few points unclear in the abstract.

1) Summer and winter cycles of moisture addition and depletion are mentioned but no information is given about the underlying processes leading to these cycles

Response: We will change the sentence "We identified two major cycles of rock moisture addition and depletion, one in the summer, and the other in the winter." into "We identified two major cycles of rock moisture addition and depletion, one in the summer which is affected by air vapor concentration and condensation, and the other in the winter which is caused by freezing-thawing".

2) LSTMs are used to predict soil moisture. But, it is not clear whether different LSTM's are used for summer and winter and whether different input variables are used for the two different seasons.

Response: We only qualitatively described the cause of fluctuating rock moisture in the winter. The winter period was not used to train the LSTM and was not predicted by the LSTM model.

3) vapor concentration and the difference in dew point temperature and wall temperature are informative input variables to predict rock moisture. It is mentioned that they improved the prediction performance but it is unclear compared to what the predictions were improved.

Response: Sorry for our unclear expression. The improvement in prediction is reflected by the decreased MAE and RMSE, which are labeled in Figure 6 in the manuscript (Figure S3 in the current file).

[Figure]

Figure S3. The predicted and measured rock water content obtained by two schemes with different input variables. (a) Scheme #1 using four directly measured variables; (b) Scheme #2 using two calculated variables controlling vapor condensation. Also shown are NSE of the whole time series, MAE and RMSE of three different stages.

4) What are 'multi-diurnal fluctuations'?

Response: It should be "multiday fluctuations". Sorry for our misuse of the word.

5) Causal factors of rock moisture fluctuations were identified. But, it is not clear which fluctuations the authors are referring to. Fluctuations in a season or diurnal fluctuations? Can explaining the moisture fluctuation in a season be used to explain the diurnal fluctuations?

Response: Sorry for our unclear expression. Diurnal rock moisture fluctuations should have causal factors different from seasonal fluctuations. The term "multi-diurnal fluctuations" in the abstract should be changed to "multiday fluctuations".

Introduction:

I was confused by the word use 'cave'. It can be either an underground camber or it can be a cavity in a face of a cliff. Looking at the pictures later, it seems that the latter definition is the one applicable here. This difference is important to understand where air temperature, humidity is measured or defined. Furthermore, are air humidity and temperature measured in these caves or in the free atmosphere at some distance from the caves? I suppose that there will also be an exchange of moisture and sensible heat between the free air and the wall of the cliff and that the humidity temperature in the cavities will be different from that in the free atmosphere. I think you need to elaborate on this and explain better where air humidity, atmospheric conditions, etc. are measured and how those measurements are influenced or influence the rock moisture.

Ln 57: You want to investigate the relation between atmospheric conditions and the rock moisture content in caves. But where are the atmospheric conditions measured? In the caves or above the soil surface in the free atmosphere?

Ln 70-75: Make clear here whether you are referring to air temperature and humidity in the caves.

Response: According to the UNESCO World Heritage Centre (https://whc.unesco.org/en/list/1039), "cave" is used officially to represent the "cavity" with rock carvings.

We totally agree that the humidity and temperature in the caves are different from that in the free atmosphere outside the caves, and there is exchange of moisture and

sensible heat between the free air and the wall of the cliff. In our study, air humidity and temperature are measured next to the wall of the cliff (the pink points shown in Figure 2 in the manuscript). We will describe this more clearly in the revision.

Study site and methods:

I think that a conceptual figure that shows the 'different forms of water' and the different 'sources of these forms' would be helpful.

Response: Thanks for your suggestion. We will try to plot a conceptual figure in the revision.

Figure 3 and Figure 4 should be made conform with each other. An output layer is missing in figure 4. The notation in the text should match with the notation used in figures 3 and 4. The equations in the text were mixed up.

Response: Thanks for pointing out the problems. We will fix these problems in the revision.

Ln 155: Dimensions of the weight matrices should be given.

Response: Thanks for your suggestion. We will give the dimensions of the weight matrices in the revision.

Ln 185: equations for mean absolute error and root mean square error are not correct.

Response: Thanks for pointing out the problem. We will fix the errors in the revision.

Results:

Rock moisture increases with increasing vapor concentration, also when wall temperature is higher than the dew point temperature. Is that caused by the fact that the water potential in the rock is much lower than the water potential of free water? The dew point gives the temperatures at which vapor starts condensing on a flat surface. But, wouldn't vapor also condense in a dry porous medium when the temperature is

above the dew point temperature but the vapor concentration is above the equilibrium vapor concentration in the free air space that is in contact with the water in the porous medium? This equilibrium vapor concentration would be calculated from the water potential in the porous medium using the Kelvin equation.

Response: We totally agree with you. Following your approach, we have calculated the equilibrium vapor concentration in the porous medium from the water potential and wall temperature by the Kelvin equation.

    As shown in Figure S4, the period with air vapor concentration being higher than the equilibrium vapor concentration in the porous medium corresponds to the period when condensation occurs. We will also add Figure S4 into the revision.

[Figure]

Figure S4. The equilibrium vapor concentration in the porous medium.

**Reference:**

Gao, S., Huang, Y. F. Zhang, S., Han, J. C., Wang, G. Q., Zhang, M. X. and Lin, Q. S.: Short-term runoff prediction with GRU and LSTM networks without requiring time step optimization during sample generation, J. Hydrol., 589, 125188, https://doi.org/10.1016/j.jhydrol.2020.125188, 2020.

Kratzert, F., Klotz, D., Brenner, C., Schulz, K. and Herrnegger, M.:Rainfall–runoff modelling using Long Short-Term Memory (LSTM) networks, Hydrol. Earth Syst. Sci, 22, 6005–6022, https://doi.org/10.5194/hess-22-6005-2018, 2018.

Fang, Z. C., Wang, Y., Peng, L. and Hong H. Y.: Predicting flood susceptibility using LSTM neural networks, J. Hydrol., 594, 125734, https://doi.org/10.1016/j.jhydrol.2020.125734, 2021.

---

## Author Comment (AC3)

**Response to Reviewer #3**

This paper investigates the dynamics of rock moisture in a sandstone cave and describes the relationship between rock moisture, surface and air temperature, and relative humidity and precipitation. The main and innovative methodology used by the authors is an FDR soil moisture probe to derive the dynamics of rock moisture. For the analysis of the influencing variables, the authors use 4 months of hourly data from one year as training data for an LSTM model to predict the hourly water content in the rock during 4 months in the second year. The most important results are that the water content in the rock is subject to seasonal fluctuations. It increases in seasons with high humidity and high temperatures. Using the SHAP values shows that precipitation is not used as a predictor variable by the LSTM. However, the LSTM has high prediction scores (NSE) based on measurements of humidity and temperature variables, which is consistent with theoretical principles.

(I am not a specialist in rock moisture dynamics or LSTM but understand a bit about modeling and ML techniques and am confident about my knowledge in field soil moisture measurements and soil vapor adsorption.)

In my opinion, the topic is very interesting and I agree with the authors that it is necessary to make progress in this field and to explore new measurement methods. Also, their results make a lot of sense from a theoretical point of view. Unfortunately, however, I have doubts about the methodology and data analysis which need to be addressed first before minor points could be discussed.

I suggest major revision for the following reasons:
TDR soil moisture probes have apparently been used already by other authors in this field to measure rock moisture dynamics (l. 48f, l.100f) but the authors state they "attempt to use the FDR for monitoring rock moisture in the field for the first time" (l.

56). Since this methodology is used in a new application setting, I strongly recommend giving more details about the sensor installation procedure. (l.97 ff). For me, it is not clear if the sensor is placed into holes drilled into the wall to be in close proximity/enclosed by the wall. This should be additionally added as a picture in Figure 2. I consider this an essential piece of information since it is known in the soil science community that contact between the sensor and (soil) parent material is crucial to obtain valid records of the (soil) moisture level.

Response: We totally agree that some previous studies (Mollo and Greco, 2011; Rempe and Dietrich, 2018; Sakaki and Rajaram, 2005) have used the TDR technique to monitor rock moisture, which have been cited in our manuscript. The difference between FDR and TDR has been mentioned in our manuscript. To the authors' knowledge, although the FDR technique has been widely used to monitor soil moisture, there is no previous application of FDR for rock moisture. Would you please tell us some previous application of FDR in rock moisture? Many thanks!

Following your suggestion, we will add the following picture of the installed sensor in Figure 2. We totally agree that the contact between the sensor and parent material is crucial to obtain accurate measurement. During the installation procedure, we use fine sand as infilling to make sure that the FDR sensor is in close proximity to the hole. We will add the details in the revision.

[Figure]

Figure S1. The FDR sensor in Cave #9.

Additionally, I feel more information is needed on the sensor's measurement sensitivity as well as the temperature sensitivity of the sensor. Although it is stated that FDR is

"less influenced by temperature" (l. 55f) compared to TDR I suggest adding more information about the temperature effect. Ideally, a sensor calibration to temperature would be cited or performed under controlled conditions to exclude the possibility that Temperature has a dominant influence on the FDR reading, particularly because it varies only between 0.010 and 0.030 (1-3% volumetric moisture).

Response: We have conducted experiments by installing a FDR sensor into a rock specimen, which is placed outdoors. We do find the fluctuating temperature induced by sunshine caused significant temperature effect, showing increasing FDR signals with temperature.

In the current study, the FDR sensor is installed inside a cave. As shown in Figure 5, the wall temperature is quite stable in the summer. Because the sensor is close to the wall, we believe that the sensor temperature is close to the wall temperature. Figure S2 shows that in the periods with obvious rock moisture addition, there is no obvious correlation between apparent rock water content and wall temperature in either summer or winter. We will discuss why the temperature effect can be neglected in the revision.

[Figure]

Figure S2. Some periods with obvious rock moisture addition. The rock moisture addition does not accompany increasing temperature.

The application of LSTM (as a Machine learning technique) is currently a hot topic and widespread in the scientific community but I am not sure if the use of this method is really necessary for the data analysis in this study. The results obtained from the LSTM have NSE scores as high as 0.958 and 0.97. From what I have been taught, such high scores usually need to be investigated very cautiously. Therefore, I wanted to clarify again that the LSTM was trained on a different period than the one shown in Figure 6? Please also provide information on the scores for the training data.

Response: Yes, the LSTM was trained on a different period than the one shown in Figure 6. We use the summer of 2020 as the training period and the summer of 2021 for testing. As shown in Figure S3, in the training period, the NSE of the Scheme #1 is as high as 0.988 and the NSE of the Scheme #2 is as high as 0.990.

[Figure]

Figure S3. The result of training period of (a) scheme #1 and (b) scheme #2.

Another possibility for achieving in such high NSE would be, that the predictor variables are very highly correlated with the dependent variable (rock moisture), so the model has a very "easy task". I quickly reviewed the data from the .pdf document provided by the authors on Zenodo (only 226 observations because I copied them out

of the .pdf format) and it looks like all variables are highly correlated and correlations are all significant. These results would question i) the need to use such a sophisticated method (LSTM) and ii) more generally, the validity of the FDR sensor data as used in this study (because of a possible temperature effect that is superimposed on the rock moisture measurement). I suggest to at least include additionally to the LSTM statistically more simple (and easier to interpret) scores of the relationship between all predictor variables and the rock water content.

Response: We agree that rock water content is highly correlated with humidity and air temperature. Table 1 shows the covariance between rock moisture and normalized atmospheric conditions in 2020 and 2021. The results show that each variable has a positive correlation with rock moisture. Vapor concentration ($C_v$) has the largest covariance while rainfall (P) has the lowest covariance. This is consistent with the feature importance obtained by the SHAP method. Although we did not predict rock moisture by using other classical statistical methods, we believe that the LSTM network, which is an emerging deep learning approach, has its unique advantages over other classical methods when it is combined with the SHAP method.

Table S1. The covariance between rock moisture and atmospheric conditions

|      | $T_a$ | RH    | P     | $T_w$ | $C_v$ | $T_d$-$T_w$ |
|------|-------|-------|-------|-------|-------|-------------|
| 2020 | 0.030 | 0.039 | 0.001 | 0.034 | 0.047 | 0.030       |
| 2021 | 0.043 | 0.034 | 0.001 | 0.036 | 0.044 | 0.022       |

We have tried to predict rock moisture using a RNN model with the same schemes. Although the results predicted by the RNN model is also acceptable (Figure S4), the resulting NSE is not as high as that obtained by the LSTM model. This is because the LSTM is a variant model that addresses the limitations of traditional RNNs by improving their ability to handle long-term dependencies. Therefore, we believe that the LSTM network as a deep learning model has its unique advantages.

The possible temperature effect has been discussed in our reply to the previous comment.

[Figure]

Figure S4. The predicted result of RNN by (a) scheme #1 and (b) scheme #2.

Therefore, before continuing the review process I strongly recommend to I) clarify the installation procedure of the FDR probe, ii) clarify the sensitivity of the instrument to temperature, and if the sensitivity is constant in time and iii) check for their whole data set, how strongly the variables are correlated to figure out if the use of LSTM is even necessary, or if the effect of the variables on the FDR reading is direct.

Response:

i) The installation procedure of the FDR probe has been clarified in our response and will be introduced in our revision.

ii) The possible temperature effect has been clarified in our response and will be introduced in our revision.

iii) The correlation between the variables has been revealed in our response and will be introduced in our revision.

iiii) The advantages of the LSTM network over the RNN network has been compared in our response and will be introduced in our revision.

---

## Author Response (AR1)

**Cover Letter**

Dear Editor,

Thank you for providing detailed reviews of our manuscript ***Physics-informed machine learning for understanding rock moisture dynamics in a sandstone cave***. Following the suggestions of the three reviewers, we have conducted major revision to the manuscript.

A detailed response is attached in this file. If there is any problem with the revision, please let me know.

Thank you for your consideration.

Best regards,

Xiao-Wei Jiang

Professor of Hydrology

E-mail: jxw@cugb.edu.cn

**Response to Editor**

Public justification (visible to the public if the article is accepted and published):

Dear Authors.

Thank you for submitting your responses to the three reviewers' comments. The experimental study on the causes of rock moisture in a cultural heritage site in China, based on field measurements, is an interesting contribution to this special issue. However, the manuscript contains several inconsistencies, and the quality of English language use sometimes makes the text difficult to understand. Despite this, I appreciated the authors' detailed and relevant responses to the reviewers' comments.

Response: Thanks for your positive assessment on our study. We have incorporated the reviewers' comments in the revision. We also fixed some grammatical errors in the revision. We believe that the quality of the current version of the manuscript is much better the original version.

While I appreciate the effort the authors have put into this manuscript, I believe that significant improvements are needed before it can meet the standards required for publication in a high-quality journal like HESS. If the authors are willing to make major revisions to address the issues I've identified, I would be open to reconsidering my assessment of the manuscript. For this sake, the authors must:

- Improve the manuscript by addressing all the comments and suggestions made by the reviewers. While this may require extensive revisions, I do think the authors have presented relevant replies to the reviewers; as so, all these details, corrections, and improvements must be integrated in the manuscript, so the manuscript can be enriched and elevated to a higher standard;

Response: Thanks for your suggestion. We have incorporated almost all of the reviewers' comments into the revision. Our response to the reviewers is attached.

- Enhance the quality of the English language used in the manuscript. There are several grammatical and linguistic errors that need to be addressed. For instance, Reviewer #1 has pointed out an error in the usage of 'multi-diurnal fluctuations'. To express changes that occur within a single day, it is appropriate to use the term 'intraday fluctuations'. Conversely, if changes occur over several days, the term 'interday fluctuations' would be more suitable. These is just one examples among many, and it is essential to carefully review the manuscript to ensure that all such errors are corrected.

Response: Thanks for pointing out our problems. Following your suggestion, we have used the term "intraday fluctuations" and "interday fluctuations" in the revision. We have fixed some grammatical errors in the revision.

As so, we look forward to receiving the revised version of your manuscript!

**Response to Reviewers**

Reviewer #1

This paper investigates the dynamics of rock moisture in a cultural heritage site in China. Based on an analysis of soil moisture sensor measurements, the main outcome of the study is that wetting of the rocks during the summer period is mainly caused by water absorption from the water vapor in the atmosphere. Also a wetting and drying cycle is observed during winter due to freezing and thawing. The interpretation of the results is based on sensitivity analyses of a trained LSTM and an LSTM trained on a set of available direct measurements is compared with an LSTM that is trained on variables that are derived from measurements and that are more directly related to condensation processes such as the wall temperature and dew point temperature. The latter LSTM is called a physics-informed LSTM. The LSTMs are trained on a dataset of one year and are then used to predict the other year.

The obtained insights are interesting and could also be of relevance for the management, protection of the heritage site. I would propose that the authors also give some ideas on how these insights could be used for these purposes.

Response: In this study, the vapor concentration in caves instead of rainfall infiltration is identified as the major controlling factor of rock moisture, which is responsible for weathering. Therefore, rock weathering could be alleviated by reducing water vapor condensation, which would be useful for protection of heritage sites.

There are a few general questions that need to be addressed.

1) The paper is based on a 2-years time series of rock moisture measurements by only one single sensor. What is the value of such a single sensor time series? Can the results be transferred to other locations? I can imagine that one does not want to disturb such heritage sites with a lot of sensors but wouldn't it be possible to find a few locations where sensor measurements would not cause a disturbance? Or would you expect a big difference when you place the sensors in a sand layer that you put in

thermal contact with the wall so that the wall temperature of the sand is similar to that of the rock? Is a 2-years period sufficiently long? I think these are questions that need to be addressed and discussed in the paper.

Response: (1) Thanks for your understanding. We were not permitted to install a lot of sensors to disturb the heritage site in the Yungang Grottes. By installing one FDR sensor in Cave #9, which was the first trial of using the FDR technique to monitor rock moisture, we identified the source of rock moisture from air vapor. The aim of the current paper is to report our monitoring scheme and to reveal sources of rock moisture. Although we only have one sensor, we believe that this does not undermine the effectiveness of the monitoring scheme and the understanding of mechanisms controlling rock moisture fluctuations in caves.

In fact, to test the universality of this recognition, in July 2022, we implemented rock moisture monitoring in another cave, Cave #4. The rock moisture measured in Cave #4 and Cave #9 during 21 July and 21 September 2022 are shown in Fig. S1. The similar patterns of rock moisture addition and depletion in response to vapor concentration in the two caves confirm our recognition on mechanisms controlling rock moisture fluctuations in caves. Unfortunately, the duration of monitoring in Cave #4 is short, which is not suitable to be included into the current manuscript.

[Figure]

Figure S1. Rock moisture and atmospheric conditions measured in Cave #4 and Cave #9 in the summer of 2022.

(2) As we shown in Figure S1, rock moisture in summer of 2022 is also controlled by vapor concentration. However, rock moisture addition and depletion in summer of

2022 are not as significant as those in the year 2021. Moreover, there was no obvious occurrence of water droplets in the caves in summer of 2022. Because we had good photos showing water droplets in the caves in summer of 2021 (Figure 1b in the manuscript), we believe that the year of 2021 is suitable for analysis and prediction.

In fact, we have monitoring data since 2019. After comparing prediction by using one year (2020) and two years (2019 and 2020) for training, we find the predicted results differ little (Figure S2). Because the aim of the current paper is to report our monitoring scheme and reveal sources of rock moisture, we prefer not to include the data of 2019.

[Figure]

Figure S2. The comparison of prediction results by using one year (2020) and two years (2019 and 2020) for training.

2) The description of the LSTMs is not clear and the reason for also presenting the RNN is not clear since it is not used in the paper.

Response: Thanks for your comments. We have added more detailed description of the LSTMs in the revision.

RNN is a recurrent neural network that is mainly used for modeling sequence data. A limitation of common RNNs is that they cannot capture the long time dependence of sequences, that is, some earlier historical data are ignored when conducting predictions on long time series data. The LSTM is a variant model that improves the common RNNs for long-term dependence. This paper used a LSTM model to capture more accurate temporal correlations between time series data of rock water content (RWC).

In this paper, the reason for presenting common RNNs is to introduce the improvement of the LSTM model over the common RNN. We have added "The LSTM network is an improved variant of the conventional Recurrent Neural Network (RNN), which is a recurrent neural network that is mainly used for modeling sequence data. Because the LSTM network has the same fundamental framework as the conventional RNN, we first briefly introduce the structure of RNN." at the beginning of 3.1. We have also added more details of the LSTM model in 3.1.

3) The physics-informed LSTM performed a bit better than the other LSTM but the improvements are not very impressive. (1) Could you find examples where you would expect a much larger improvement? Even though the match between the measurement and LSTM is impressive, they remain a black box and it is not so clear to me (2) why the LSTMs are needed in this paper. What is their role? Doing a sensitivity analyses, you found that precipitation did not explain the rock moisture dynamics and comparing model fits of physics-informed LSTMs with non-informed LSTMs you found that absorption of vapor is the main process. But, can't this be inferred as well from time series analyses, e.g. covariance, wavelet, analyses? If the purpose is to obtain an understanding of the processes, what are the advantages of using LSTMs in comparison with other more classical methods? Could the parameters or weights that are obtained by training the LSTM be interpreted and used to explain the behavior of the system?

Response: (1) Thanks for acknowledging that our physics-informed LSTM performed better than the non-physics-informed LSTM. Unfortunately, probably because rock moisture in July 2021 is the highest in the years monitored, the physics-informed LSTM still underestimates the measurements in July 2021. Note that in the other two periods of summer 2021, one is before 10 July and the other is after 5 August, the prediction and measurements have very good match.

(2) We agree that a LSTM model capable of high precision prediction is a black box. In our study, the combination of the LSTM model and the SHAP (SHapley Additive

exPlanations) method leads to not only high accuracy of prediction, but also interpretability of predictions. Therefore, by combining with the SHAP method, the LSTM model is no longer a black box.

We totally agree that there are other methods that can be used to reveal the correlation between two time series or to predict rock moisture. Table 1 shows the covariance between rock moisture and normalized atmospheric conditions in 2020 and 2021. The results show that each variable has a positive correlation with rock moisture. Vapor concentration ($C_v$) has the largest covariance while rainfall (P) has the lowest covariance. This is consistent with the feature importance obtained by the SHAP method.

Table S1. The covariance between rock moisture and atmospheric conditions

|  | $T_a$ | RH | P | $T_w$ | $C_v$ | $T_d$-$T_w$ |
|---|---|---|---|---|---|---|
| 2020 | 0.030 | 0.039 | 0.001 | 0.034 | 0.047 | 0.030 |
| 2021 | 0.043 | 0.034 | 0.001 | 0.036 | 0.044 | 0.022 |

We have also predicted rock moisture using a conventional RNN model. The physics-informed model also performs better than the non-informed model (Figure S3). Although the results predicted by the RNN model is also acceptable, the resulting NSE is not as high as that obtained by the LSTM model. This is because the LSTM is a variant model that addresses the limitations of traditional RNNs by improving their ability to handle long-term dependencies. Therefore, we believe that the LSTM network as a deep learning model has its unique advantages, especially when it is combined with the SHAP method.

[Figure]

Figure S3. The predicted result by using the conventional RNN.
(a) scheme #1; (b) scheme #2.

4) I was confused about the winter period. Was it also used to train the LSTM? There is a discussion on the sensor measurements during the winter period but I could not find a discussion on the performance of the LSTMs for this period.

Response: Sorry for our unclear expressions.

We want to clarify that the winter period was not used to train the LSTM and was not predicted by the LSTM model. In line 235, we pointed out that "Because the period from 1 June to 1 October has the most significant trends of rock moisture addition and depletion, the hourly data during this period in the year 2020 are used to construct the training set, whereas the hourly data in the year 2021 are used to construct the test set".

Abstract:

There were a few points unclear in the abstract.

1) Summer and winter cycles of moisture addition and depletion are mentioned but no

information is given about the underlying processes leading to these cycles

Response: We have changed the sentence "We identified two major cycles of rock moisture addition and depletion, one in the summer, and the other in the winter." into "We identified two major cycles of rock moisture addition and depletion, one in summer which is affected by air vapor concentration and condensation, and the other in winter which is caused by freezing-thawing".

2) LSTMs are used to predict soil moisture. But, it is not clear whether different LSTM's are used for summer and winter and whether different input variables are used for the two different seasons.

Response: We only qualitatively described the cause of fluctuating rock moisture in the winter. The winter period was not used to train the LSTM and was not predicted by the LSTM model.

3) Vapor concentration and the difference in dew point temperature and wall temperature are informative input variables to predict rock moisture. It is mentioned that they improved the prediction performance but it is unclear compared to what the predictions were improved.

Response: Sorry for our unclear expression. The improvement in prediction is reflected by the increased NSE and decreased MAE and RMSE. We have modified the sentence into "…the predicted RWC has a better agreement with the measured RWC, with increased NSE and decreased MAE and RMSE"

4) What are 'multi-diurnal fluctuations'?

Response: Sorry for our unclear expression. Following the suggestion of the editor, we have changed it into "interday fluctuations".

5) Causal factors of rock moisture fluctuations were identified. But, it is not clear which fluctuations the authors are referring to. Fluctuations in a season or diurnal

fluctuations? Can explaining the moisture fluctuation in a season be used to explain the diurnal fluctuations?

Response: Sorry for our unclear expression. Diurnal rock moisture fluctuations should have causal factors different from seasonal fluctuations. The term "multi-diurnal fluctuations" in the abstract has been changed to "interday fluctuations".

Introduction:

I was confused by the word use 'cave'. It can be either an underground camber or it can be a cavity in a face of a cliff. Looking at the pictures later, it seems that the latter definition is the one applicable here. This difference is important to understand where air temperature, humidity is measured or defined. Furthermore, are air humidity and temperature measured in these caves or in the free atmosphere at some distance from the caves? I suppose that there will also be an exchange of moisture and sensible heat between the free air and the wall of the cliff and that the humidity temperature in the cavities will be different from that in the free atmosphere. I think you need to elaborate on this and explain better where air humidity, atmospheric conditions, etc. are measured and how those measurements are influenced or influence the rock moisture.

Ln 57: You want to investigate the relation between atmospheric conditions and the rock moisture content in caves. But where are the atmospheric conditions measured? In the caves or above the soil surface in the free atmosphere?

Ln 70-75: Make clear here whether you are referring to air temperature and humidity in the caves.

Response: According to the UNESCO World Heritage Centre (https://whc.unesco.org/en/list/1039), "cave" is used officially to represent the "cavity" with rock carvings in the Yungang Grottoes.

We totally agree that the humidity and temperature in the caves are different from that in the free atmosphere outside the caves, and there is exchange of moisture and sensible heat between the free air and the wall of the cliff. In our study, air humidity

and temperature are measured next to the wall of the cliff (the pink points shown in Figure 2 in the manuscript).

In the revision, we have added the following sentence "Air temperature (Ta) and air relative humidity (RH) are simultaneously monitored near the monitoring site of rock moisture (Fig. 2). The wall temperature is also monitored to analyze whether the wall meets the condition for condensation. Moreover, hourly precipitation is available from a meteorological station outside the cave."

Study site and methods:

I think that a conceptual figure that shows the 'different forms of water' and the different 'sources of these forms' would be helpful.

Response: Thanks for your suggestion. Here we mentioned two forms of water here, one is water droplet, and the other is rock moisture. Because there are only two forms of water, we think a better way is to make the sentence clear. We have modified it into "Water in the form of either water droplets or rock moisture is responsible for weathering of the statues, however, the sources of the two forms of water remain controversial."

Figure 3 and Figure 4 should be made conform with each other. An output layer is missing in figure 4. The notation in the text should match with the notation used in figures 3 and 4. The equations in the text were mixed up.

Response: Thank you so much for your valuable comments! Figure 3 shows a RNN model containing an input layer, a hidden layer, and an output layer, where the hidden layer consists of RNN cells. The original Figure 4 shows a LSTM cell, which is a unit in the hidden layer of the LSTM model. We have revised Figure 4 to show a complete LSTM model. Please see the revised Figure 4.

Ln 155: Dimensions of the weight matrices should be given.

Response: Thanks for your suggestion. We have given the dimensions of the weight

matrices in the revision. The sentence is "The dimensions of $W_{fx}$, $W_{ix}$, $W_{ox}$ and $W_{cx}$ are $D×M$, and the dimensions of $W_{fh}$, $W_{ih}$, $W_{oh}$ and $W_{ch}$ are $M×M$, where $D$ is the number of input features and $M$ is the number of hidden units in the LSTM layer."

Ln 185: Equations for mean absolute error and root mean square error are not correct.

Response: Thanks for pointing out the problem. We have fixed the errors in the revision.

Results:

Rock moisture increases with increasing vapor concentration, also when wall temperature is higher than the dew point temperature. Is that caused by the fact that the water potential in the rock is much lower than the water potential of free water? The dew point gives the temperatures at which vapor starts condensing on a flat surface. But, wouldn't vapor also condense in a dry porous medium when the temperature is above the dew point temperature but the vapor concentration is above the equilibrium vapor concentration in the free air space that is in contact with the water in the porous medium? This equilibrium vapor concentration would be calculated from the water potential in the porous medium using the Kelvin equation.

Response: We totally agree with you. Unfortunately, as pointed out by Reviewer #2, it is difficult to calibrate the measured rock water content. In the current study, we use the apparent rock water content to establish the relationship between rock moisture additions/depletions and atmospheric conditions.

By assuming that the apparent and actual rock water contents are close enough, following your approach, we have calculated the equilibrium vapor concentration in the porous medium from the water potential and wall temperature by the Kelvin equation. As shown in Figure S4, the period with air vapor concentration being higher than the equilibrium vapor concentration in the porous medium corresponds to the period when condensation occurs.

In the current study, because we have not figured out a way to calibrate the rock

water content yet, we prefer not to include this result in the revision. We believe that your suggestion would be useful for our future studies.

[Figure]

Figure S4. The equilibrium vapor concentration in the porous medium.

Reviewer #2

This is an excellent paper on rock moisture in sandstone caves of a cultural heritage site. The authors point out that rock moisture is a hidden component of the terrestrial hydrologic cycle" and I fully agree. The authors use a FDR moisture sensor for measuring rock water content (RWC) over approx. 2 years and meteorological measurements to explain the observed RWC fluctuations. They use a machine learning algorithm to assess the explanatory potential of each meteorological factor and find that air humidity and dewpoint temperature explain a large part of the observed fluctuations. RWC is extremely well predicted from input variables (e.g. Fig. 6). This is a very interesting and unique result. I recommend acceptance after very minor revision.

(I must note that I'm a specialist on rock moisture but I don't have the expertise to assess the quality of the machine learning approach.)

Response: Thanks for your positive assessment on our study.

Specific comments:

L104-106: I can confirm from my own work that on-site calibration of RWC measurements is difficult and that the relationship is linear in approximation.

Response: It is true that on-site calibration of RWC measurements is difficult and we just use the apparent RWC in the current study.

L222-224: "The decreasing low liquid water content induced by freezing indicates that rock moisture in spring and autumn belongs to movable bound water that could be responsible for chemical weathering." I cannot follow this reasoning. Consider to leave this out, or explain better why you think this.

Response: Thanks for pointing out the problem. We have deleted this unclear sentence. In the revision, we have added the following sentences: "The increased RWC before freezing indicates that there is movable water in winter even if the RWC is very low" and "The movable water could be responsible for chemical weathering".

L231, L253, L261: I'm not sure if precipitation can in fact be discarded as an input factor of RWC. There cannot be a direct correlation between P and RWC as the up and down of single precipitation events does not reach the rock surfaces in the caves. If the water seeped through the rock, it would reach the surfaces (a) with a considerable time lag, (b) in a greatly smoothed temporal course. Have you tried to feed in e.g. a running average of precipitation with a temporal shift of several days (or weeks) into your model?

Response: We want to clarify that if infiltrating precipitation can reach the caves, the lagged response of rock moisture to precipitation can be identified by the LSTM model.

In fact, there is no fractures in our monitoring site and we believe that infiltrating precipitation cannot reach the site. In our study, we use the SHAP model to confirm that the rock moisture has no correlation with precipitation infiltration, but is controlled mainly by air humidity.

Editorial comments:

L121: Write tan together (without spaces, otherwise it looks like four different variables). I my pdf, several other formula seem to be slightly disturbed.

Response: Sorry for the incorrect forms of the formula due to transformation of a docx file into a pdf file. We have fixed this problem in the revision.

L206-208: A lot of repetitive words in this section: Simplify to: "... indicating that this slight change is a result of the fluctuating rock water content. Therefore, we believe that the FDR readings reflect the actual change of water content in the rock."

Response: Thanks for your suggestion. We have replaced the sentence in the revision.

L210-211: Simplify to: "This pattern of fluctuation is a direct consequence of freezing-thawing, ..."

Response: Thanks for your suggestion. We have replaced it in the revision.

L250-252: Put numbers in brackets behind the parameters: "The mean absolute SHAP values are in descending order: air relative humidity (0.0087), air temperature (0.0032)..." and so on

Response: Thanks for your suggestion. We have corrected the sentence by following your suggestion.

Entire Paper: The term rock water content is quite frequent; consider to use the abbreviation RWC.

Response: Thanks for your suggestion. We have changed all the term "rock water content" into the abbreviation RWC in the revision.

Reviewer #3

This paper investigates the dynamics of rock moisture in a sandstone cave and describes the relationship between rock moisture, surface and air temperature, and relative humidity and precipitation. The main and innovative methodology used by the authors is an FDR soil moisture probe to derive the dynamics of rock moisture. For the analysis of the influencing variables, the authors use 4 months of hourly data from one year as training data for an LSTM model to predict the hourly water content in the rock during 4 months in the second year. The most important results are that the water content in the rock is subject to seasonal fluctuations. It increases in seasons with high humidity and high temperatures. Using the SHAP values shows that precipitation is not used as a predictor variable by the LSTM. However, the LSTM has high prediction scores (NSE) based on measurements of humidity and temperature variables, which is consistent with theoretical principles.

(I am not a specialist in rock moisture dynamics or LSTM but understand a bit about modeling and ML techniques and am confident about my knowledge in field soil moisture measurements and soil vapor adsorption.)

In my opinion, the topic is very interesting and I agree with the authors that it is necessary to make progress in this field and to explore new measurement methods. Also, their results make a lot of sense from a theoretical point of view. Unfortunately, however, I have doubts about the methodology and data analysis which need to be addressed first before minor points could be discussed.

I suggest major revision for the following reasons:
TDR soil moisture probes have apparently been used already by other authors in this field to measure rock moisture dynamics (l. 48f, l.100f) but the authors state they "attempt to use the FDR for monitoring rock moisture in the field for the first time" (l. 56). Since this methodology is used in a new application setting, I strongly

recommend giving more details about the sensor installation procedure. (l.97 ff). For me, it is not clear if the sensor is placed into holes drilled into the wall to be in close proximity/enclosed by the wall. This should be additionally added as a picture in Figure 2. I consider this an essential piece of information since it is known in the soil science community that contact between the sensor and (soil) parent material is crucial to obtain valid records of the (soil) moisture level.

Response: We totally agree that some previous studies have used the TDR technique to monitor rock moisture. To our knowledge, although the FDR technique has been widely used to monitor soil moisture, there is no previous application of FDR for rock moisture. To avoid confusion, we have deleted the sentence "we attempt to use the FDR for monitoring rock moisture in the field for the first time" in the revision. The reason of selecting FDR is stated as "Because FDR sensors have the advantage of small in volume (the length is less than 60 mm), for minimizing disturbance to rocks in heritage sites, we attempt to use the FDR sensor to monitor rock moisture in a cave with stone carvings."

Following your suggestion, we have added the following photo of the installed sensor in Figure 2 in the manuscript. We totally agree that the contact between the sensor and parent material is crucial to obtain accurate measurement. During the installation procedure, we use fine sand as infilling to make sure that the FDR sensor is in close proximity to the hole. We have added more details in Section 2.2 of the revision.

[Figure]

Figure S5. The installed FDR sensor in Cave #9.

Additionally, I feel more information is needed on the sensor's measurement

sensitivity as well as the temperature sensitivity of the sensor. Although it is stated that FDR is "less influenced by temperature" (l. 55f) compared to TDR I suggest adding more information about the temperature effect. Ideally, a sensor calibration to temperature would be cited or performed under controlled conditions to exclude the possibility that Temperature has a dominant influence on the FDR reading, particularly because it varies only between 0.010 and 0.030 (1-3% volumetric moisture).

Response: Thanks for pointing out the potential problems of the FDR sensors. We conducted experiments by installing a FDR sensor into a rock specimen, which is placed outdoors. We do find the fluctuating temperature induced by sunshine could cause significant temperature effect, i.e., showing increasing FDR signals with temperature.

In our study, the FDR sensor is installed inside a cave. As shown in Fig. S6, the daily mean wall temperature is quite stable from 11 July to 10 August and from 15 December to 23 December (with limited diurnal fluctuations), but there are obvious trends of rock moisture addition. Therefore, we are sure that the fluctuating FDR readings reflect the rock moisture additions and depletions. In the revision, we have added Fig. S6 and discussed why the temperature effect can be neglected.

[Figure]

Figure S6. Two typical periods with obvious rock moisture addition. The rock moisture addition does not accompany increasing temperature.

The application of LSTM (as a Machine learning technique) is currently a hot topic and widespread in the scientific community but I am not sure if the use of this method is really necessary for the data analysis in this study. The results obtained from the LSTM have NSE scores as high as 0.958 and 0.97. From what I have been taught, such high scores usually need to be investigated very cautiously. Therefore, I wanted to clarify again that the LSTM was trained on a different period than the one shown in Figure 6? Please also provide information on the scores for the training data.

Response: Yes, the LSTM was trained on a different period than the one shown in Figure 6 in the manuscript. We use the summer of 2020 as the training period and the summer of 2021 for testing.

As shown in Figure S7, in the training period, the NSE of the Scheme #1 is as high as 0.988 and the NSE of the Scheme #2 is as high as 0.990.

[Figure]

Figure S7. The result of observed and predicted RWC in the training period.
(a) scheme #1; (b) scheme #2.

Another possibility for achieving in such high NSE would be, that the predictor variables are very highly correlated with the dependent variable (rock moisture), so

the model has a very "easy task". I quickly reviewed the data from the .pdf document provided by the authors on Zenodo (only 226 observations because I copied them out of the .pdf format) and it looks like all variables are highly correlated and correlations are all significant. These results would question i) the need to use such a sophisticated method (LSTM) and ii) more generally, the validity of the FDR sensor data as used in this study (because of a possible temperature effect that is superimposed on the rock moisture measurement). I suggest to at least include additionally to the LSTM statistically more simple (and easier to interpret) scores of the relationship between all predictor variables and the rock water content.

Response: First of all, the possible temperature effect has been discussed in our reply to the previous comment.

We agree that rock water content is highly correlated with humidity and air temperature. Table S2 shows the covariance between rock moisture and normalized atmospheric conditions in 2020 and 2021. The results show that each variable has a positive correlation with rock moisture. Vapor concentration ($C_v$) has the largest covariance while rainfall (P) has the lowest covariance. This is consistent with the feature importance obtained by the SHAP method.

Table S2. The covariance between rock moisture and atmospheric conditions

|      | $T_a$ | RH    | P     | $T_w$ | $C_v$ | $T_d$-$T_w$ |
|------|-------|-------|-------|-------|-------|-------------|
| 2020 | 0.030 | 0.039 | 0.001 | 0.034 | 0.047 | 0.030       |
| 2021 | 0.043 | 0.034 | 0.001 | 0.036 | 0.044 | 0.022       |

We have also predicted rock moisture by using a RNN model. Although the results predicted by the RNN model is also acceptable (Figure S8), the resulting NSE is not as high as that obtained by the LSTM model. This is because the LSTM is a variant model that addresses the limitations of conventional RNNs by improving their ability to handle long-term dependencies. Therefore, we believe that the LSTM network as a deep learning model has its unique advantages.

[Figure]

Figure S8. The predicted results by using the RNN.
(a) scheme #1; (b) scheme #2.

Therefore, before continuing the review process I strongly recommend to I) clarify the installation procedure of the FDR probe, ii) clarify the sensitivity of the instrument to temperature, and if the sensitivity is constant in time and iii) check for their whole data set, how strongly the variables are correlated to figure out if the use of LSTM is even necessary, or if the effect of the variables on the FDR reading is direct.

Response:

i) The installation procedure of the FDR probe has been clarified in our response and in our revision.

ii) The possible temperature effect has been clarified in our response and in our revision.

iii) The correlation between the variables has been revealed in our response.

iiii) The advantages of the LSTM network over the RNN network has been compared in our response.

---

## Referee Report (RR1)

**Reviewer #3 comment round 2: Physics-informed machine learning for understanding rock moisture dynamics in a sandstone cave; Kai-Gao Ouyang et al.; https://doi.org/10.5194/hess-2022-403**

I appreciate the considerable efforts made by the authors to enhance the manuscript since the previous round of reviews, and their diligence in addressing all the questions raised.
However, I regret to say that I remain unconvinced by their response to the central point of my critique, namely the impact of sensor sensitivity to temperature. This remains a point of uncertainty for me.

By presenting two instances of two-week periods with elevated FDR readings despite no observable temperature changes (within the figure), the authors draw the conclusion that the sensor's sensitivity can be considered insignificant. First of all, I suggest changing this figure so that the scale of the y-axis for temperature reflects the range of the data which is shown in the figure, currently the difference between minimum and maximum value is too large.

I would like to support my argument again by referring to the state of literature. It is well known that the electrical permittivity of soils (for which the TDR and FDR sensors have been developed) is influenced "by temperature (Roth et al., 1990; Wraith and Or, 1999; Owen et al., 2002; Rosenbaum et al., 2011), (soil) texture (Ponizovsky et al., 1999) and organisation of thin water film layers (Wang and Schmugge, 1980)". (p 648, Jackisch et al. 2018). Jackisch et al. (2020) also show in their Figure A3 that most soil moisture probes have a high correlation with temperature.

Even with soils, it is considered ideal to calibrate the sensor under controlled conditions for the respective application. As this study relates to a single sensor in a new application and calibration under controlled conditions requires few resources (sensor, stone, scale, thermometer) I would recommend the authors to perform this calibration. In my eyes, this simple measurement would essentially improve their study.

Without this calibration the potential temperature effect needs to be a central point of the discussion and needs to be more prominently addressed in every section of the manuscript, including the abstract, introduction, and the conclusions.

My strong recommendation about this aspect is based on the fact that the range measured by the sensor reported in this study is very low (0.010-0.030 over the whole period of observations) while the change in temperature is not negligible (-15 to 14 °C according to Fig. 5; Tsurface 8.4 – 13.6 °C ; Tair 10.9 – 21.4 °C between June and September 2021 from the downloaded data)

Additionally, I was suggesting in the first review round to the authors to include a more simple method to measure the relationship between the predictor variables than the LSTM to give a baseline to which the performance of the more sophisticated method can be compared. The authors provided a table with the covariance between the signal of the dielectric permittivity (interpreted as RWC) and the normalized atmospheric conditions in 2020 and 2021.
Although this response goes into the right direction, in this context the correlation coefficient is more meaningful than the covariance. Additionally, since my question refers to the necessity of the LSTM, I argue that the same period of time should be considered (June – September). I copy pasted some data out of the pdf provied by the  authors (covering summer 2021, please provide .csv ot .txt format in the

future when providing data) and calculated the correlation coefficient between predictor variables and the response variable (RWC) (see below). Although the LSTM outperforms a linear model in this study the linear model would already be doing a very good job (RWC ~ T_surface, $r^2$ = 0.71). Additionally, the strong correlations between RWC and both temperature measurements (0.84 & 0.82) underline the point that a proper validation (under controlled conditions) that the sensor is not mainly influenced by temperature would be recommendable (or this aspect needs to be highlighted).

I think the authors can still substantially improve the quality of their research and this manuscript by addressing the above mentioned points.

[Figure]

correlation chart of all provided variables for the above shown period (pearson correlation coefficient with degree of significance):

[Figure]

RWC = rock water content, T_s = wall surface temperature, T_a = air temperature cave, RH = relative humidity, cv = vapor concentration, Tdew = dewpoint temperature, PP = Precipitation.

---

## Author Response (AR2)

**Cover Letter**

Dear Editor,

Thank you for providing detailed reviews of our manuscript *Physics-informed machine learning for understanding rock moisture dynamics in a sandstone cave*. The reviewer raised concerns about our reasoning regarding the temperature effect. In the response, we provide a data set of **outdoor FDR measurements** with large diurnal variations of temperature to show how to identify the temperature effect, and give evidence of negligible temperature effect of **indoor FDR measurements in our study site inside a cave**. A detailed response to the reviewer is attached in this file.

If there is any problem with the revision, please let me know.

Thank you for your consideration.

Best regards,

Xiao-Wei Jiang

Professor of Hydrology

E-mail: jxw@cugb.edu.cn

**Response to Editor**

Dear Authors,

Thank you for submitting your responses to the two reviewers' comments. I greatly appreciate the significant effort you have made to improve this manuscript. I have also noticed a significant improvement in the quality of the English language used throughout the manuscript.

While I am pleased with the authors' detailed and relevant responses to the reviewers' comments, there are still some important issues that need to be addressed. Reviewer #2 has provided a comprehensive report, raising concerns about the authors' reasoning regarding the temperature effect on the measurement signal. The authors claim that this effect is negligible, but Reviewer #2 challenges this reasoning.

Therefore, I kindly request the authors to specifically address this issue raised by Reviewer #2.

Thank you for your attention to this matter.

Response: Thanks for your positive assessment on our study. We have provided detailed explanations and evidence of negligible temperature effect of FDR measurements in our study site inside a cave. We have also revised our manuscript.

**Response to Reviewer#2**

I appreciate the considerable efforts made by the authors to enhance the manuscript since the previous round of reviews, and their diligence in addressing all the questions raised.

However, I regret to say that I remain unconvinced by their response to the central point of my critique, namely the impact of sensor sensitivity to temperature. This remains a point of uncertainty for me.

By presenting two instances of two-week periods with elevated FDR readings despite no observable temperature changes (within the figure), the authors draw the conclusion that the sensor's sensitivity can be considered insignificant. First of all, I suggest changing this figure so that the scale of the y-axis for temperature reflects the range of the data which is shown in the figure, currently the difference between minimum and maximum value is too large.

I would like to support my argument again by referring to the state of literature. It is well known that the electrical permittivity of soils (for which the TDR and FDR sensors have been developed) is influenced "by temperature (Roth et al, 1990; Wraith and Or, 1999; Owen et al, 2002; Rosenbaum et al, 2011), (soil) texture (Ponizovsky et al, 1999) and organisation of thin water film layers (Wang and Schmugge, 1980)". (p 648, Jackisch et al 2018). Jackisch et al (2020) also show in their Figure A3 that most soil moisture probes have a high correlation with temperature.

Even with soils, it is considered ideal to calibrate the sensor under controlled conditions for the respective application. As this study relates to a single sensor in a new application and calibration under controlled conditions requires few resources (sensor, stone, scale, thermometer) I would recommend the authors to perform this calibration. In my eyes, this simple measurement would essentially improve their study. Without this calibration the potential temperature effect needs to be a central point of the discussion and needs to be more prominently addressed in every section of the manuscript, including the abstract, introduction, and the conclusions.

My strong recommendation about this aspect is based on the fact that the range measured by the sensor reported in this study is very low (0.010-0.030 over the whole period of

observations) while the change in temperature is not negligible (-15 to 14 °C according to Fig. 5; Tsurface 8.4 – 13.6 °C; Tair 10.9 – 21.4 °C between June and September 2021 from the downloaded data).

Additionally, I was suggesting in the first review round to the authors to include a more simple method to measure the relationship between the predictor variables than the LSTM to give a baseline to which the performance of the more sophisticated method can be compared. The authors provided a table with the covariance between the signal of the dielectric permittivity (interpreted as RWC) and the normalized atmospheric conditions in 2020 and 2021. Although this response goes into the right direction, in this context the correlation coefficient is more meaningful than the covariance. Additionally, since my question refers to the necessity of the LSTM, I argue that the same period of time should be considered (June – September). I copy pasted some data out of the pdf provided by the authors (covering summer 2021, please provide .csv ot .txt format in the future when providing data) and calculated the correlation coefficient between predictor variables and the response variable (RWC) (see below). Although the LSTM outperforms a linear model in this study, the linear model would already be doing a very good job (RWC ~ T_surface, r2 = 0.71).

Additionally, the strong correlations between RWC and both temperature measurements (0.84 & 0.82) underline the point that a proper validation (under controlled conditions) that the sensor is not mainly influenced by temperature would be recommendable (or this aspect needs to be highlighted).

I think the authors can still substantially improve the quality of their research and this manuscript by addressing the above mentioned points.

Response: We summarize that the reviewer still has three concerns of our study. The first is whether there is temperature effect associated with the FDR measurements, the second is the necessity of sensor calibration, and the third is the necessity of the LSTM model.

**1. On the temperature effect**

We agree that the FDR measurement is sensitive to temperature, especially when the temperature spans a large range. In the references you listed, the temperature variations

are really large. For example, the temperature ranges between 5 and 65°C in Wraith and Or (1999), between 0 and 70°C in Owen et al (2002) and between 5 and 40°C in Rosenbaum et al (2011). We acknowledge that the FDR readings which have temperature effect cannot represent the rock water content, however, we have evidence to prove that temperature effect of FDR sensors inside the caves in this study is negligible.

To demonstrate that the sensor installed in Cave #9 free from sunshine exposure has negligible temperature effect, we first compare our latest measurements carried out in an outdoor sandstone which is exposed to sunshine and in the indoor sandstone inside Cave #9. In the outdoor sandstone, as a result of the large temperature variations induced by solar radiation, there is obvious diurnal variation of FDR signal (Fig. S1a); in the indoor sandstone, as a result of the small temperature variation, there is not obvious diurnal variation of FDR signal, especially in September and October (Fig. S1b).

[Figure]

Figure S1. The fluctuating RWC and $T_w$ in August through October of 2022. (a) Outdoor sandstone and (b) Indoor sandstone inside Cave #9. Note that a and b have the same scale.

For the FDR sensor installed in outdoor sandstone exposed to sunshine, there is a linear correlation between RWC and $T_w$, with $R^2$ being as high as 0.92 and 0.996 in September and October (Fig. S2a), respectively. However, this does not mean that the correlation between RWC and $T_w$ is enough to reveal the temperature effect. By defining $\Delta RWC = RWC_t - RWC_{t-1}$ and $\Delta Tw = T_t - T_{t-1}$, we find a high correlation between $\Delta RWC$ and $\Delta T$, with $R^2$ being as high as 0.96 and 0.98 in September and October, respectively (Fig. S2b). In August, $R^2$ of correlation between $\Delta RWC$ and $\Delta T$ is also as high as 0.67. Therefore, outdoor FDR measurement has temperature effect, which can be revealed by using a plot of $\Delta RWC$ versus $\Delta T$.

[Figure]

Figure S2. Plots of RWC versus $T_w$ (a) and of hourly variation in RWC, $\Delta RWC$, versus hourly variation in $T_w$, $\Delta T_w$, (b) in the outdoor sandstone in three months.

However, in our manuscript, the sensor was installed inside Cave #9 that is not exposed to sunshine. In the revision, we use plots of $\Delta RWC$ versus $\Delta T$ in four months (Fig. S3b) to exclude the possible occurrence of temperature effect. Although there is

correlation between RWC and $T_w$ in September (with $R^2$ equals 0.78, Fig. S3a2), there is no obvious correlation between $\Delta$RWC and $\Delta$T (Fig. S3b2), indicating that an instantaneous change in temperature does not cause an instantaneous change in FDR signal. Moreover, there is no correlation between $\Delta$RWC and $\Delta$T is all other months. Therefore, we want to clarify that the correlation between RWC and $T_w$ cannot be used as an indicator of the temperature effect.

In the revision, we use plots of $\Delta$RWC versus $\Delta T_w$ to illustrate that the temperature effect is negligible.

[Figure]

Figure S3. (a) Plots of RWC versus $T_w$ in Cave #9; (b) Plots of $\Delta$RWC versus $\Delta T_w$ in Cave #9.

**2. On the necessity of sensor calibration**

We totally agree that sensor calibration is important. However, as pointed out by Reviewer #2, calibrating rock water content measurements in the field remains a challenge so far.

We have tried to calibrate the rock water content of rock samples in the laboratory. By weighing rock samples, we can establish a linear relationship between FDR readings and rock water content. However, in the current study, the FDR sensor is installed into a rock wall, the weight of which is impossible to be weighed. Moreover, we cannot collect a large rock sample to install a FDR sensor and conduct sensor calibration.

Because the possible occurrence of temperature effect has been excluded, we believe that direct reading of FDR measurements can be used to understand the dynamics of rock moisture.

**3. On the necessity of the LSTM model**

As indicated in the title of our manuscript "Physics-informed machine learning for understanding rock moisture dynamics in a sandstone cave", the aim of our study is to improve understanding of causes of rock moisture dynamics with the aid of a machine learning model. The cause-and-effect relationship between rock water content and vapor concentration is the precondition of high precision prediction.

We totally agree with the reviewer that a simple linear model can do a good enough job to predict rock moisture. However, it is impossible to have high $R^2$ by using other traditional models. In our study, the LSTM model leads to high $R^2$ equaling 0.985 and 0.996 for schemes #1 and #2, respectively, illustrating that the LSTM model outperforms a linear model. In our opinion, a high $R^2$ close to 1 predicted by the LSTM model is a robust indicator showing that the fluctuating rock moisture is caused by water vapor condensation.